# Global Burden of Rare Cancers: Insights from GLOBOCAN 2022 Estimates

**DOI:** 10.3390/cancers17101721

**Published:** 2025-05-21

**Authors:** Mohammed Elmadani, Simon Klara, Mohammed Mustafa, Evans Kasmai Kiptulon, Mate Orsolya

**Affiliations:** 1Doctoral School of Health Sciences, Faculty of Health Sciences, University of Pecs, 7622 Pecs, Hungary; klara.simon@etk.pte.hu (S.K.); mohammad.qadr@tiu.edu.iq (M.M.); evans.kasmai.kiptulon@pte.hu (E.K.K.); 2Department of Epidemiology, Faculty of Public Health, University of El Imam El Mahdi, Kosti 27711, Sudan; 3Jamhuriya Research Center, Jamhuriya University of Science and Technology, Mogadishu 2526, Somalia

**Keywords:** rare cancers, incidence and mortality, GLOBOCAN 2022, bladder cancer, non-Hodgkin lymphoma, leukemia, pancreatic cancer, esophageal cancer

## Abstract

Rare cancers, though individually uncommon, collectively account for a significant portion of global cancer cases and deaths. Understanding their incidence and mortality patterns is crucial for improving early detection and treatment strategies. This study analyzed global data from 2022 to identify the most common rare cancers and their regional distribution. The findings show that cancers such as bladder cancer, leukemia, and non-Hodgkin lymphoma have high incidence rates, while pancreatic and esophageal cancers are associated with poor survival. Geographic differences highlight disparities in healthcare access and cancer management across regions. By identifying these patterns, this research provides valuable insights for policymakers and healthcare professionals to enhance cancer screening, diagnosis, and treatment strategies. Addressing these challenges could help improve outcomes for patients with rare cancers and guide future research on reducing the burden of these diseases worldwide.

## 1. Introduction

Cancer remains a leading cause of morbidity and mortality worldwide, with significant disparities in incidence, treatment, and survival outcomes [1,2,3]. While considerable research has focused on common malignancies, rare cancers represent a significant but often overlooked global health burden [4,5]. Rare cancers, defined as malignancies with an incidence of fewer than 6 cases per 100,000 individuals annually, collectively represent a significant portion of the global cancer burden [5,6,7]. Despite their rarity, these cancers account for approximately 25% of all cancer diagnoses worldwide [8,9,10,11]. In Europe alone, about 5.1 million people are living with rare cancers, with more than 650,000 new cases diagnosed each year [10,12]. Similarly, in the United States, rare cancers constitute approximately 20% of all cancer diagnoses [12].

Despite their relative infrequency, rare cancers pose unique challenges, including late diagnoses, limited research funding, and a lack of standardized treatment protocols [7,13]. Patients with rare cancers often face challenges such as delayed diagnoses and limited treatment options, leading to poorer outcomes [14,15]. According to the International Agency for Research on Cancer (IARC), which is a specialized agency of the World Health Organization (WHO), the five-year survival rate for rare cancers is about 49%, significantly lower than the 63% observed for more common cancers [9,10]. This disparity underscores the need for enhanced research efforts, improved diagnostic methods, and the development of effective therapies tailored to these uncommon malignancies.

The epidemiological burden of rare cancers varies across geographical regions, influenced by genetic, environmental, and healthcare system factors [16,17]. Studies suggest that the cumulative incidence of rare cancers in Europe is comparable to that of more common malignancies, highlighting the necessity for enhanced surveillance and research [9]. Furthermore, while advancements in histomolecular classifications have improved understanding, rare cancers remain underrepresented in clinical trials, limiting treatment options and innovation [13,18,19].

Disparities in healthcare infrastructure further exacerbate the burden of rare cancers, particularly in low- and middle-income countries where access to specialized oncology care is limited [20,21]. The recent establishment of global initiatives, such as the Global Society of Rare Genitourinary Tumors, seeks to address these disparities by promoting collaborative research and knowledge-sharing [11,22].

Given the challenges associated with rare cancers, there is an urgent need for targeted public health strategies, including improved diagnostic methodologies, centralized treatment facilities, and increased investment in research [13,23,24]. Addressing these gaps will be critical in reducing mortality and improving the quality of life for patients affected by rare malignancies. This paper aims to provide a comprehensive analysis of the global burden of rare cancers based on GLOBOCAN 2022 estimates, focusing on incidence and mortality rates, diagnostic and treatment challenges, and emerging strategies to improve outcomes.

## 2. Materials and Methods

### 2.1. Data Source

This study utilized data from the GLOBOCAN 2022 database (https://gco.iarc.who.int (accessed on 17 March 2025)), the most recent global cancer estimation resource compiled by the International Agency for Research on Cancer (IARC) in collaboration with the World Health Organization (WHO). The analysis included age-standardized incidence and mortality rates (ASRs) per 100,000 person-years, allowing for comparisons between populations while adjusting for differences in age structures [9].

The data were stratified by continent (Africa, Latin America and the Caribbean, Northern America, Europe, Oceania, and Asia) and encompassed all age groups and genders. The number of new rare cancer cases and deaths were extracted for 24 types classified under the International Classification of Diseases, Tenth Edition (ICD-10), all of which had an incidence rate of fewer than 6 cases per 100,000 individuals.

Cancer incidence refers to the number of newly diagnosed cases within a given population over a specific time frame, whereas cancer mortality represents the number of deaths attributed to cancer within the same period. Incidence data were primarily collected through national or regional population-based cancer registries, while cause-specific mortality statistics were sourced from national vital registration systems [25].

### 2.2. Data Extraction and Variables

Based on GLOBOCAN 2022 estimates, the rare cancers included in this study were as follows: (1) head and neck: lip, oral cavity (C00-06), salivary glands (C07-08), oropharynx (C09-10), nasopharynx (C11), hypopharynx (C12-13); (2) digestive system: esophagus (C15), gallbladder (C23), pancreas (C25); (3) respiratory and skin: larynx (C32), melanoma (C43), mesothelioma (C45), Kaposi sarcoma (C46); (4) genitourinary system: vulva (C51), vagina (C52), penis (C60), testis (C62), kidney (C64), bladder (C67); and (5) central nervous system and hematological: brain and central nervous system (C70-72), Hodgkin lymphoma (C81), non-Hodgkin lymphoma (C82-86, C88), multiple myeloma (C90), leukemia (C91-95).

### 2.3. Data Analysis

We present a comparative analysis of incidence and mortality across continents for the five most common rare cancers worldwide. The GCO platform provides facilities for data tabulation and graphical visualization at the global, regional, and national levels, stratified by cancer type, sex, and age. GLOBOCAN estimates are derived from each country’s best available cancer incidence and mortality data sources, with methodologies relying on short-term projections and modelled mortality-to-incidence ratios where applicable. The methods used to compile the 2022 estimates are largely based on previous approaches, ensuring consistency and comparability.

As of 2022, the GLOBOCAN database provides estimates for 36 cancer types across 185 countries and 21 regions, incorporating current, historical, and projected data on cancer incidence, mortality, and prevalence by site and sex [26]. Among these, 23 cancer types are classified as rare cancers, defined by a global incidence of fewer than 6 cases per 100,000 population.

Finally, projections of rare cancer prevalence and rare-cancer-related deaths through 2050 were presented based on different cancer sites and regional comparisons across continents. These estimates were derived from demographic projections, assuming constant cancer incidence and mortality rates over time. The results are expressed as the expected percentage increases in the number of rare cancer cases and related deaths.

### 2.4. Ethical Considerations

Ethical approval was not required since all data used were anonymized and aggregated at the national level. However, this study adhered to the Declaration of Helsinki and ethical guidelines for secondary data analysis.

## 3. Results

In 2022, rare cancers constituted a significant global health burden, accounting for 5,347,784 new cases and 2,959,369 deaths, representing 26.7% of all new cancer cases and 30% of all cancer-related deaths worldwide. Among these, bladder cancer had the highest incidence rate, with an age-standardized rate (ASR) of 5.58 per 100,000, contributing to 614,298 new cases and 220,596 deaths. Other highly prevalent rare cancers included non-Hodgkin lymphoma (NHL) with an ASR of 5.57 per 100,000 (553,389 cases and 250,679 deaths), leukemia with an ASR of 5.26 (487,294 cases and 305,405 deaths), esophageal cancer (511,054 cases and 445,391 deaths), pancreatic cancer (510,992 cases and 467,409 deaths), and kidney cancer (434,840 cases and 155,953 deaths). Cancers of the lip and oral cavity (ASR: 4.00), brain and central nervous system (ASR: 3.47), and melanoma of the skin (ASR: 3.22) also posed significant health challenges. Notably, while melanoma had a relatively lower mortality ASR (0.53 per 100,000), cancers of the brain and central nervous system exhibited a high fatality burden, with 248,500 deaths (ASR: 3.47) from 321,731 cases (ASR: 2.59). Figure 1 shows the age-standardized rates (ASRs) for all rare cancer estimations.

The distribution of rare cancers varied across continents Table 1, reflecting disparities in healthcare access, early detection, and treatment outcomes. Bladder cancer had the highest incidence in Europe (ASR: 12.04, 224,777 cases), followed by North America (ASR: 11.02, 95,546 cases), while Asia, despite its large population, had a lower incidence rate (ASR: 3.38, 215,755 cases). Europe also had the highest mortality rate (ASR: 2.98, 70,383 deaths), whereas Asia recorded the highest total number of deaths (92,510 deaths, ASR: 1.35). Similarly, leukemia had the highest incidence in North America (ASR: 11.21, 71,667 cases) and Oceania (ASR: 10.19, 6669 cases), but Asia recorded the largest absolute number of cases (227,206 cases, ASR: 4.46) and the highest total deaths (158,144 deaths, ASR: 2.9). Non-Hodgkin lymphoma (NHL) showed a similar pattern, with North America having the highest incidence rate (ASR: 12.49, 87,466 cases), while Asia had the highest total cases (235,442 cases, ASR: 4.02) and deaths (121,525 deaths, ASR: 2.0). Africa had the highest NHL mortality ASR (3.26 per 100,000).

Esophageal cancer posed a significant challenge in Asia, which bore the highest burden (ASR: 6.22, 382,892 cases), with 329,803 deaths (ASR: 5.28), reflecting limited early detection and treatment. In contrast, Africa exhibited a high mortality-to-incidence ratio (29,965 cases, 28,276 deaths, ASR: 3.48), indicating poor survival outcomes. Pancreatic cancer had the highest incidence in Europe (ASR: 7.98, 146,477 cases) and North America (ASR: 8.45, 67,089 cases), while Asia recorded the highest total cases (232,537 cases) but had a lower ASR (3.64).

Figure 2 illustrates the projected percentage increase in rare cancer cases among males from 2022 to 2050. All cancer types analyzed are expected to show a rising trend, though the magnitude of increase varies significantly across cancer sites. The steepest increases are projected for mesothelioma and bladder cancer, each expected to surpass a 100% increase by 2050. More moderate but still substantial increases are observed for penile, esophageal, central nervous system (CNS) tumors, and multiple myeloma. In contrast, cancers such as testicular cancer, Hodgkin lymphoma, leukemia, and multiple myeloma are projected to rise at a relatively slower rate, with estimated increases between 20% and 40% over the same period. Overall, this upward trajectory highlights a concerning and widespread rise in the incidence of rare cancers among males through 2050.

Similarly, Figure 3 presents the projected percentage increase in the number of rare cancer cases among females from 2022 to 2050. Again, all cancer types are expected to increase, although the magnitude varies. The steepest increases are projected for pancreatic, bladder, and vulvar cancers, each anticipated to rise by 80–100% by 2050. In contrast, cancers such as Kaposi sarcoma, Hodgkin lymphoma, and nasopharyngeal cancer are expected to increase more moderately, with estimated rises between 30% and 50%. As with males, the overall upward trend indicates a growing burden of rare cancers among females over the coming decades.

Figure 4 presents global estimates of the percentage increase in new rare cancer deaths among females aged 0–85+ from 2022 to 2050. In 2022, death rates for all listed rare cancers were relatively low, generally under 10%. However, projections for 2050 show a substantial increase in mortality. Notably, bladder and pancreatic cancers are expected to exhibit the most significant increases, surpassing 100%. Other cancers—including lip, oral cavity, nasopharynx, hypopharynx, larynx, esophagus, gallbladder, skin melanoma, vulva, vagina, kidney, Hodgkin lymphoma, non-Hodgkin lymphoma, and leukemia—show increases ranging from 60% to 90%. Kaposi sarcoma shows a more modest increase, projected at less than 30%.

Figure 5 displays global projections of new rare cancer deaths among males aged 0–85+ from 2022 to 2050. As in females, estimated death percentages for all listed rare cancers in 2022 were generally below 10%. By 2050, substantial increases are expected, particularly for bladder cancer, which is projected to rise by more than 120%. Other cancers—including lip, oral cavity, nasopharynx, salivary glands, esophagus, larynx, gallbladder, skin melanoma, penis, kidney, Hodgkin lymphoma, non-Hodgkin lymphoma, multiple myeloma, and leukemia—are also projected to see marked increases. Testicular cancer and Kaposi sarcoma are expected to experience more moderate growth in mortality.

Figure 6 compares the projected burden of rare cancer deaths across continents from 2022 to 2050. Among males, the global number of rare cancer deaths is projected to increase from 1.85 million in 2022 to 3.45 million in 2050—an increase of 86.0%. Regionally, Africa is expected to experience the highest relative increase (143.2%, from 0.12 to 0.30 million), followed by Latin America and the Caribbean (92.6%, from 0.13 to 0.25 million) and Oceania (93.8%, from 0.02 to 0.03 million). Asia will bear the highest absolute burden, with deaths increasing from 1.05 to 1.95 million (+86.8%). Europe and Northern America are expected to see more moderate increases of 38.5% (0.39 to 0.54 million) and 69.9% (0.14 to 0.25 million), respectively.

Among females, global deaths from rare cancers are projected to rise from 1.11 million in 2022 to 2.09 million in 2050—an increase of 88.9%. Again, Africa is expected to experience the highest relative increase (140.7%, from 0.09 to 0.22 million), followed by Oceania (+99.3%, from 0.01 to 0.02 million), Latin America and the Caribbean (+97.2%, from 0.09 to 0.17 million), and Asia (+97.6%, from 0.57 to 1.13 million). North America and Europe are projected to experience smaller increases of 62.5% (0.09 to 0.15 million) and 32.0% (0.25 to 0.33 million), respectively. These projections emphasize the growing global burden of rare cancer mortality, with disproportionately higher increases in low- and middle-income regions.

Figure 7 shows the estimated number of new cancer cases by 2050 compared to 2022 for both males and females across world regions, along with the corresponding percentage increases. The global cancer burden is projected to rise significantly for both sexes: 75.3% for males (from 3.39 to 5.94 million new cases) and 76.0% for females (from 1.96 to 3.45 million). Africa is expected to experience the most pronounced increases: 133.9% for males (from 0.19 to 0.44 million) and 130.7% for females (from 0.14 to 0.32 million). Latin America and the Caribbean also show substantial increases of 79.4% (males) and 83.2% (females). Asia, which already carries the highest burden, is projected to experience increases of 74.0% for males (from 1.64 to 2.86 million) and 82.2% for females (from 0.88 to 1.60 million). Other regions such as Europe, Northern America, and Oceania are expected to experience more moderate increases ranging from 44.8% to 74.8%. These projections underscore the urgent need for region-specific strategies in cancer prevention and control to address the growing disparities in cancer incidence.

Despite the lower incidence of rare cancers compared to common cancers, they account for a disproportionately high mortality rate, underscoring challenges in early diagnosis and effective treatment. The geographic variations in incidence and mortality highlight inequities in healthcare access and cancer management strategies worldwide. Addressing these disparities through improved cancer screening, early diagnosis, and equitable access to advanced treatment options is essential to reducing the burden of rare cancers globally.

## 4. Discussion

The findings on the global burden of rare cancers in 2022 highlight their significant impact on public health, with rare cancers accounting for approximately 26.7% of all new cancer cases and 30% of all cancer-related deaths. This substantial contribution emphasizes the need for enhanced research efforts, improved early detection strategies, and innovative treatment approaches to mitigate their effects. Rare cancers, though individually uncommon, collectively represent a significant proportion of the global cancer burden. They are defined by their low incidence rates, typically fewer than six cases per 100,000 people annually in the European Union [27,28,29]. Despite their rarity, rare cancers account for approximately 20–25% of all cancer diagnoses worldwide, making them a critical public health concern [30,31,32,33]. In Europe alone, over 500,000 new cases of rare cancers are diagnosed each year, and more than 4 million people are living with a rare cancer diagnosis [34,35].

Among the various rare cancers, bladder cancer had the highest incidence rate, followed closely by non-Hodgkin lymphoma and leukemia. These cancers not only present with high incidence rates but also contribute significantly to cancer-related mortality. For instance, leukemia, despite being prevalent worldwide, remains a major cause of cancer-related deaths, highlighting the need for improved therapeutic interventions. Other rare cancers, such as esophageal, pancreatic, and kidney cancers, also contribute significantly to global mortality rates, underscoring the aggressive nature of these malignancies and the challenges in their management. The burden of rare cancers is further underscored by their poor survival outcomes. The five-year relative survival rate for rare cancers is significantly lower than that for common cancers, standing at approximately 47–48% compared to 63–65% for common cancers [27,35]. This disparity is attributed to challenges in diagnosis, limited therapeutic options, and the lack of specialized care for these cancers [34,36].

Additionally, brain and central nervous system cancers, along with melanoma, illustrate disparities in mortality rates. While melanoma exhibits a relatively low mortality rate, brain and central nervous system cancers pose a substantial burden due to their high fatality rates. This discrepancy calls for targeted research efforts in neuro-oncology and skin cancer prevention. The high mortality rate associated with rare cancers is exacerbated by their often-advanced stage at diagnosis. For example, 59% of rare solid tumors are diagnosed at regional or distant stages, compared to 45% for common cancers [5]. This late diagnosis, coupled with the limited availability of effective treatments, contributes to the poorer survival outcomes observed in rare cancer patients [27].

Furthermore, pancreatic cancer, despite being the twelfth most common cancer globally, ranks seventh in cancer-related mortality, reflecting its aggressive nature and poor prognosis [37]. In 2019, there were approximately 530,297 incident cases and 531,107 deaths worldwide, with age-standardized incidence and mortality rates of 6.6 per 100,000 person-years each [38]. The disease burden, measured in disability-adjusted life years (DALYs), reached 11,549,016, highlighting its significant impact on global health [38]. Over the past three decades, the incidence and mortality rates of pancreatic cancer have steadily increased, with the age-standardized incidence rate (ASIR) and age-standardized mortality rate (ASMR) rising annually by 0.83% and 0.77%, respectively [38,39]. Projections suggest that this trend will persist, with a further rise in both incidence and mortality expected by 2044 [40]. Pancreatic cancer, notorious for its poor prognosis, displays considerable regional variations. Europe and North America have the highest incidence and mortality ASRs, reflecting the aggressive nature of this disease, while Asia, despite recording the highest absolute number of cases, has a lower ASR due to its large population. The persistently high mortality rates worldwide indicate an urgent need for advancements in early detection and treatment strategies. Geographical variations further reveal that Europe and North America continue to experience the highest burden of pancreatic cancer, with the European region identified as a hotspot for incidence and mortality [41,42]. In East Asia, particularly China, a rapid increase in pancreatic cancer cases has been observed, with incidence rates nearly doubling from 1990 to 2019 [43]. In contrast, Southeast Asia and Africa report relatively lower incidence and mortality, though these regions are witnessing rising cases due to increasing exposure to risk factors such as smoking and metabolic disorders [41,44]. Several risk factors contribute to the rising global burden of pancreatic cancer, with smoking accounting for approximately 21.4% of related deaths [38,45]. Metabolic risks, including high fasting plasma glucose and elevated body mass index (BMI), are also significant contributors, particularly in developing regions [45,46]. The aging population further exacerbates the burden, with individuals aged 55 and older accounting for an increasing proportion of cases, rising from 79.01% in 1990 to 84.41% in 2019 in Asia alone [44]. Additionally, males exhibit higher incidence and mortality rates compared to females, largely due to higher smoking and alcohol consumption rates, along with metabolic risk differences [38,44]. Notably, early-onset pancreatic cancer (EOPC), diagnosed in individuals aged 15–49 years, has emerged as a growing concern. Between 1990 and 2019, the age-standardized incidence and mortality rates for EOPC increased by 46.9% and 44.6%, respectively, with East Asia carrying the highest burden [40,47]. This trend is largely attributed to risk factors such as tobacco smoking, obesity, and high fasting plasma glucose levels [47]. Looking ahead, the global burden of pancreatic cancer is projected to rise significantly due to demographic shifts and the increasing prevalence of risk factors. By 2044, incident cases and deaths are expected to exceed 875,000 annually, underscoring the urgent need for effective prevention strategies, early detection programs, and improved therapeutic approaches to mitigate the growing impact of pancreatic cancer [40,48].

Addressing the challenges posed by rare cancers requires a multifaceted approach that includes enhanced research, early detection, and improved treatment strategies. The rarity of these cancers poses significant challenges in research and treatment, including limited patient populations for clinical trials, a lack of specialized expertise among healthcare providers, inadequate funding, and significant geographical and socioeconomic disparities in care [31,34,36]. Advances in diagnostic techniques, such as next-generation sequencing and artificial intelligence, hold promise for improving early detection and personalized treatment options [49,50,51]. Furthermore, international collaboration, decentralized clinical trials, and innovative treatment approaches, such as comprehensive genome profiling and targeted therapies, are essential for overcoming the barriers associated with rare cancers [30,50,52].

Early detection plays a pivotal role in improving outcomes for rare cancer patients. However, the low incidence of these cancers makes population-wide screening programs impractical. Instead, targeted screening strategies for high-risk populations and the use of advanced diagnostic tools, such as imaging and biomarkers, are being explored to improve early detection rates [13,50,51].

### Limitations

This study has several limitations primarily related to the use of secondary data from the GLOBOCAN 2022 database. First, the accuracy of incidence and mortality estimates is subject to the quality and completeness of national cancer registries and vital registration systems, which vary considerably across countries and regions, particularly in low- and middle-income settings. Some estimates rely on modeling and projections based on mortality-to-incidence ratios or neighboring country data, which may introduce bias or the underrepresentation of certain populations. Second, projections of rare cancer cases and deaths to 2050 assume constant incidence and mortality rates, not accounting for potential changes in healthcare systems, cancer detection, risk factor prevalence, or treatment advances over time. Third, the classification of rare cancers was based on global incidence thresholds, which may not reflect variations in rarity across different regions. Lastly, because the data are aggregated at the national level and anonymized, this analysis does not capture individual-level risk factors, disparities, or comorbidities that may influence rare cancer trends.

## 5. Conclusions

Rare cancers, though individually uncommon, collectively represent a substantial global health burden with disproportionately high mortality rates. This underscores the urgent need for improved early detection, specialized care, and targeted research. Advancements in diagnostics, such as next-generation sequencing and artificial intelligence, offer promising pathways for enhancing outcomes. Addressing disparities in access to care, particularly in low-resource settings, alongside international collaboration and policy support, is essential to improving survival and quality of life for patients with rare cancers.

## Figures and Tables

**Figure 1 cancers-17-01721-f001:**
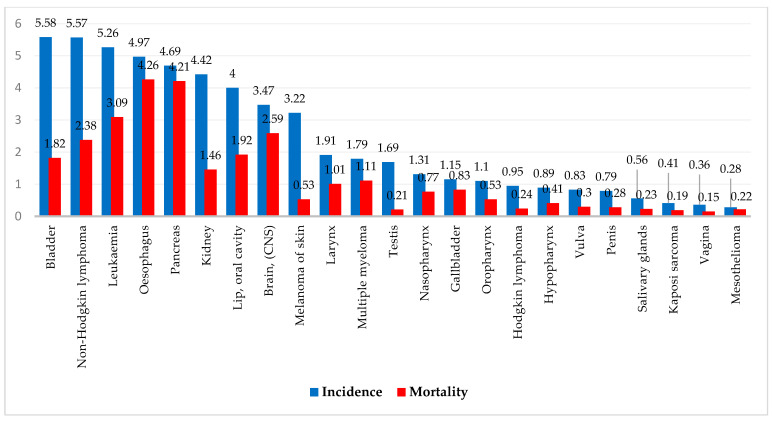
Global incidence and mortality rates (ASRs per 100,000) of rare cancers in 2022 for all ages, both sexes.

**Figure 2 cancers-17-01721-f002:**
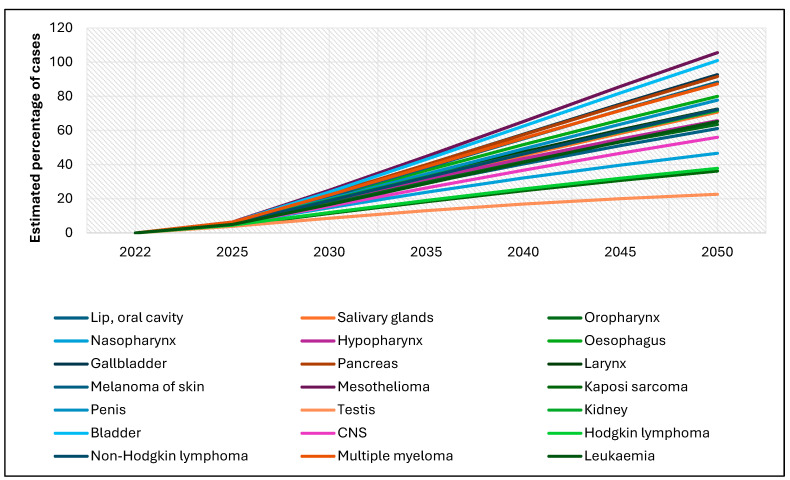
Global estimates of the percentage of new rare cancer cases among males, 2022–2050 (ages 0–85+).

**Figure 3 cancers-17-01721-f003:**
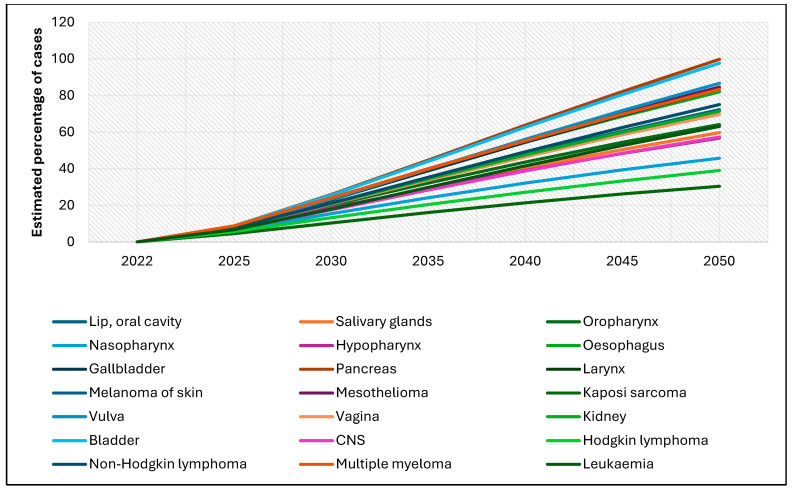
Global estimates of the percentage of new rare cancer cases among females, 2022–2050 (ages 0–85+).

**Figure 4 cancers-17-01721-f004:**
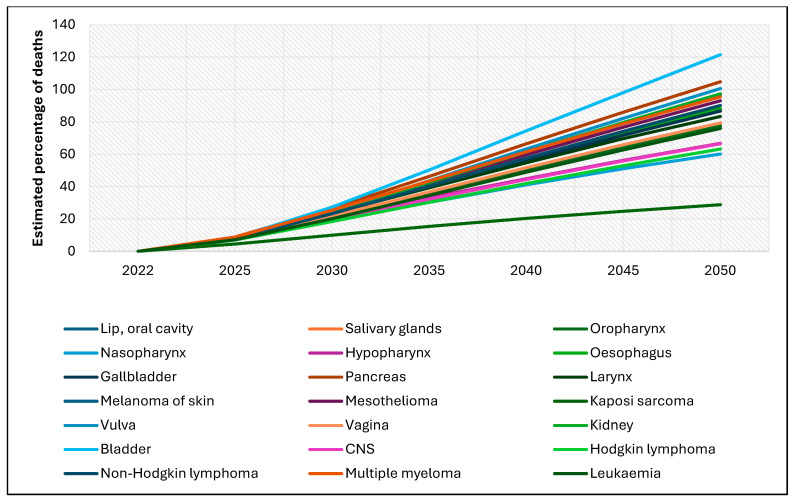
Global estimates of the percentage of new rare cancer deaths among females, 2022–2050 (ages 0–85+).

**Figure 5 cancers-17-01721-f005:**
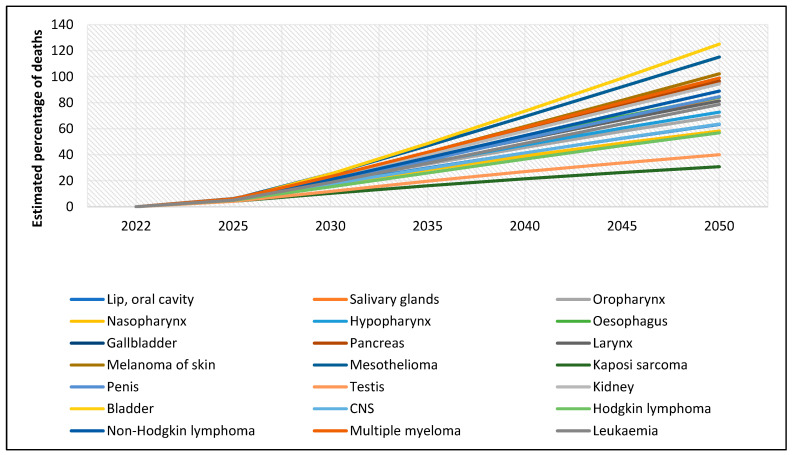
Global estimates of the percentage of new rare cancer deaths among males, 2022–2050 (ages 0–85+).

**Figure 6 cancers-17-01721-f006:**
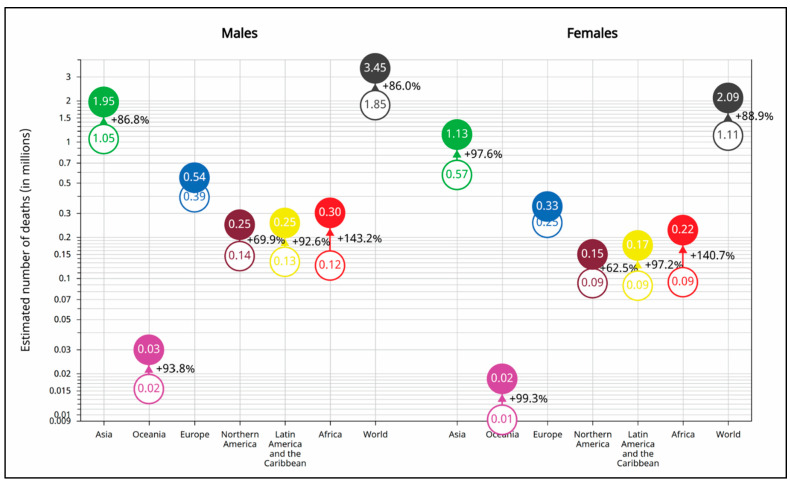
Estimated number of rare cancer deaths globally and by continent, 2022–2050, by sex (ages 0–85+).

**Figure 7 cancers-17-01721-f007:**
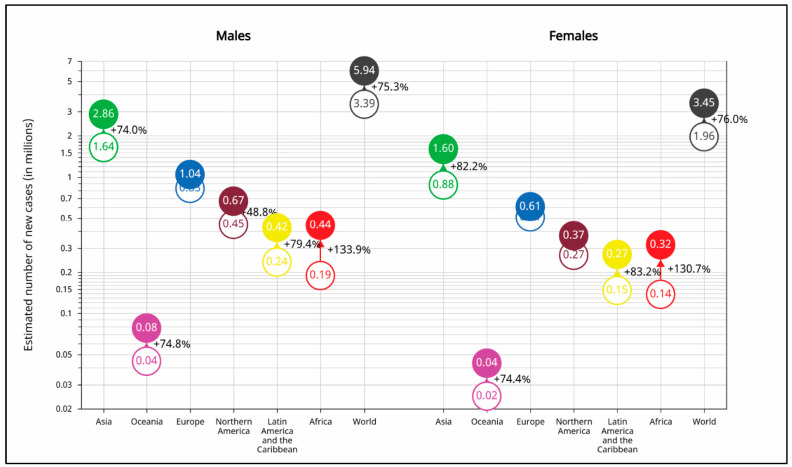
Estimated number of rare cancer cases globally and by continent, 2022–2050, by sex (ages 0–85+).

**Table 1 cancers-17-01721-t001:** Age-standardized rate (ASR, world per 100,000), total cases, and deaths for the top five rare cancers across world regions (all ages and sexes).

Cancer Type	Metric	Africa	Latin America and Caribbean	Northern America	Europe	Oceania	Asia
Bladder Cancer	Incidence ASR	4.66	3.98	11.02	12.04	6.57	3.38
	Mortality ASR	2.74	1.43	2.01	2.98	1.84	1.35
	Total Cases	37,064	35,791	95,546	224,777	5365	215,755
	Total Deaths	20,736	13,940	21,214	70,383	1813	92,510
Leukemia	Incidence ASR	3.1	5.62	11.21	7.74	10.19	4.46
	Mortality ASR	2.54	3.65	3.13	3.35	3.74	2.9
	Total Cases	32,998	41,006	71,667	107,748	6669	227,206
	Total Deaths	24,501	28,670	27,276	63,839	2975	158,144
Non-Hodgkin Lymphoma	Incidence ASR	5	5.37	12.49	8.42	11.21	4.02
	Mortality ASR	3.26	2.28	2.66	2.58	3.17	2
	Total Cases	50,497	43,128	87,466	129,338	7518	235,442
	Total Deaths	30,758	19,249	24,714	51,808	2625	121,525
Esophagus Cancer	Incidence ASR	3.64	2.38	2.88	3.31	3.28	6.22
	Mortality ASR	3.48	2.19	2.44	2.76	2.71	5.28
	Total Cases	29,965	20,366	21,888	53,513	2430	382,892
	Total Deaths	28,276	18,895	19,116	47,212	2089	329,803
Pancreatic Cancer	Incidence ASR	2.35	4.64	8.45	7.98	6.2	3.64
	Mortality ASR	2.23	4.3	6.62	7.26	5.38	3.27
	Total Cases	18,993	41,032	67,089	146,477	4864	232,537
	Total Deaths	17,770	38,319	56,044	138,644	4389	212,243

## Data Availability

Data supporting this research are available at https://gco.iarc.who.int/today/en/dataviz (accessed on 17 March 2025).

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
