# Peer review of "Global Burden of Rare Cancers: Insights from GLOBOCAN 2022 Estimates"

_cancers, 2025, doi:10.3390/cancers17101721_

Round 1
Reviewer 1 Report
Comments and Suggestions for Authors
The manuscript show interesting data. However, the format of this manuscript is really limited and larger information and data tables should be provided to support better the analysis. All information are well provided but a larger need in terms of information is needed.
The authors should also analyse the limitation of the data and possible bias.
Author Response
Comment 1:
The manuscript show interesting data. However, the format of this manuscript is really limited and larger information and data tables should be provided to support better the analysis. All information are well provided but a larger need in terms of information is needed.
Response 1:
Thank you for this helpful comment. We have expanded the data presentation by adding more figures that include detailed breakdowns of rare cancer incidence and mortality by continent, sex, and cancer type and projection of rare cancers to 2050.
Comment 2:
The authors should also analyse the limitation of the data and possible bias.
Response 2:
We appreciate this important suggestion. We have added a dedicated paragraph in the discussion section that outlines the limitations of using GLOBOCAN 2022 data, including issues related to data quality across countries, potential underreporting, and assumptions made during projections.
Best regards
Reviewer 2 Report
Comments and Suggestions for Authors
Thank you for your submission titled "Global Burden of Rare Cancers: Insights from GLOBOCAN 2022 Estimates." While the topic is undoubtedly important and timely, I must express concerns regarding the scope and methodology of your manuscript.
At present, the analysis appears to rely exclusively on data from GLOBOCAN 2022. While this is a valuable resource, it is insufficient as the sole basis for a comprehensive review of the global burden of rare cancers. There are numerous other relevant data sources and studies that could enhance the robustness and depth of your findings.
To improve the quality and reliability of the work, I recommend employing a systematic approach such as the PRISMA methodology. This would ensure transparency in the selection of sources and allow for a more exhaustive and balanced review of the existing literature.
Given these limitations, I regret to inform you that I must reject the current version of the manuscript. However, I strongly encourage you to revise and resubmit, incorporating additional sources and a more rigorous methodology.
Sincerely
Author Response
Comment 1:
At present, the analysis appears to rely exclusively on data from GLOBOCAN 2022. While this is a valuable resource, it is insufficient as the sole basis for a comprehensive review of the global burden of rare cancers. There are numerous other relevant data sources and studies that could enhance the robustness and depth of your findings.
Response 1:
We understand the concern regarding reliance on a single data source. While the aim of this study was to offer a global snapshot using the most up-to-date standardized estimates (GLOBOCAN 2022), which is compiled by the International Agency for Research on Cancer (IARC) in collaboration with the World Health Organization (WHO), this is considered one of the highest-quality and most comprehensive sources for global cancer data currently available. GLOBOCAN synthesizes incidence and mortality data from cancer registries and national vital statistics, applying validated modeling techniques where direct data is limited, to ensure consistency and comparability across countries.
We have added a dedicated paragraph in the discussion section that outlines the limitations of using GLOBOCAN 2022 data, including issues related to data quality across countries, potential underreporting, and assumptions made during projections.
Comments 2:
To improve the quality and reliability of the work, I recommend employing a systematic approach such as the PRISMA methodology. This would ensure transparency in the selection of sources and allow for a more exhaustive and balanced review of the existing literature.
Response 2:
We thank the reviewer for this valuable recommendation. As our study was designed as a global data-driven analysis using GLOBOCAN estimates rather than a literature review, the PRISMA methodology was not initially applied.
Reviewer 3 Report
Comments and Suggestions for Authors
The authors set out to analyze the global incidence and mortality of rare cancers, which is an important topic. However, the way the data is presented limits its usefulness. They only reported rates for one year (2022), without looking at how these rates have changed over time. For public health decisions, knowing how cancer rates are trending—whether they’re going up or down—is more helpful than just seeing a snapshot from a single year. Interestingly, the authors mention time comparisons in their Discussion section, which suggests they recognize this too. It would strengthen the paper if they included data from multiple years and showed trends over time. A simple measure like annual percent change could help highlight whether different types of rare cancers are becoming more or less common.
More minor comments:
- Materials and methods section could be divided into several subsections, focusing on (1) description of database; (2) data extraction and mining; (3) data analysis.
- The study limitations should be discussed.
- The Conclusions section contain many unnecessary information and could be shortened.
Author Response
Comments 1:
The authors set out to analyze the global incidence and mortality of rare cancers, which is an important topic. However, the way the data is presented limits its usefulness. They only reported rates for one year (2022), without looking at how these rates have changed over time. For public health decisions, knowing how cancer rates are trending—whether they’re going up or down—is more helpful than just seeing a snapshot from a single year.
Interestingly, the authors mention time comparisons in their Discussion section, which suggests they recognize this too. It would strengthen the paper if they included data from multiple years and showed trends over time. A simple measure like annual percent change could help highlight whether different types of rare cancers are becoming more or less common.
Response 1:
Thank you for this insightful comment. While GLOBOCAN 2022 provides only limited historical data, we have now included projected trends up to 2050 based on demographic models for selected rare cancers.
Comments 2:
- Materials and methods section could be divided into several subsections, focusing on (1) description of database; (2) data extraction and mining; (3) data analysis.
- The study limitations should be discussed.
- The Conclusions section contain many unnecessary information and could be shortened
Response 2:
- We have revised the materials and methods section accordingly, now clearly structured into four sections.
- As noted in response to reviewers 1 and 2, a dedicated paragraph on study limitations has now been included in the discussion section.
- Thank you for this helpful suggestion. We have revised and significantly shortened the conclusion section to focus on the key implications of our findings while removing unnecessary repetition.
Round 2
Reviewer 3 Report
Comments and Suggestions for Authors
Please indicate whether the projected cancer rates were calculated by you (if so, please describe the methodology) or derived from the GLOBOCAN website.
Author Response
Comment 1:
Please indicate whether the projected cancer rates were calculated by you (if so, please describe the methodology) or derived from the GLOBOCAN website.
Response 1:
Thank you for this important inquiry. The projected cancer rates presented in our study were derived from the GLOBOCAN website. We did not calculate these projections independently.
Best Regards.